# Bioinspire-Explore: Taxonomy-Driven Exploration of Biodiversity Data for Bioinspired Innovation

**DOI:** 10.3390/biomimetics9020063

**Published:** 2024-01-23

**Authors:** Adrien Saint-Sardos, Annabelle Aish, Nikolay Tchakarov, Thierry Bourgoin, Luce-Marie Petit, Jian-Sheng Sun, Régine Vignes-Lebbe

**Affiliations:** 1Centre d’Études et d’Expertises en Biomimétisme de Senlis (CEEBIOS), 62 Rue du Faubourg Saint-Martin, 60300 Senlis, France; 2Bioinspire-Museum, Museum National d’Histoire Naturelle, 57 rue Cuvier, 75005 Paris, France; annabelle.aish@mnhn.fr (A.A.);; 3Sorbonne Université, Muséum National d’Histoire Naturelle, CNRS, EPHE, Université des Antilles, Institut de Systématique Évolution Biodiversité, ISYEB, CP 48, 57 Rue Cuvier, 75005 Paris, France

**Keywords:** biodiversity, bioinspiration, data science, computer-aided biomimetics, taxonomy, open-access, NLP, biology push

## Abstract

Successful bioinspired design depends on practitioners’ access to biological data in a relevant form. Although multiple open-access biodiversity databases exist, their presentation is often adapted to life scientists, rather than bioinspired designers. In this paper, we present a new tool, “Bioinspire-Explore”, for navigating biodiversity data in order to uncover biological systems of interest for a range of sectors. Bioinspire-Explore allows users to search for inspiring biological models via taxa (species, genera, etc.) as an entry point. It provides information on a taxon’s position in the “tree of life”, its distribution and climatic niche, as well as its appearance. Bioinspire-Explore also shows users connections in the bioinspiration literature between their taxon of interest and associated biological processes, habitats, and physical measurements by way of their semantic proximity. We believe Bioinspire-Explore has the potential to become an indispensable resource for both biologists and bioinspired designers in different fields.

## 1. Introduction

Bioinspiration is a creative approach based on the observation of living systems [1]. Over the past twenty years, bioinspiration has helped solve engineering problems [2,3,4], inspire architectural programs [5], improve water management [6], support cybersafety [7,8], and even resolve in-orbit servicing challenges [9,10,11]. As impressive as these case studies are, they are based on a few dozen, perhaps a few hundred, biological models [12,13], despite more than 2 million species having been identified [14]. To reach its full potential, the field needs a greater range of “biological muses” from molecular [15] to ecosystem scales [16,17]. Advances in the fields of biology and ecology are able to provide this inspiration: we know more about living systems today than ever before. Moreover, technological developments allow us to mimic and/or incorporate nature into design in ways that were impossible even a decade ago. How can this potential be unlocked to support the ambitions of a growing number of bioinspiration organisations, startups, laboratories etc. [13,18]? Moreover, how can we draw attention to the lesser known taxonomic groups, whose diversity of form and function has much to teach us [19,20,21]? Several initiatives have been set up over the last decade(s) to support bioinspired designers’ access to biological information. Through the AskNature platform, living systems can be explored with a focus on functions, using the Biomimicry 3.8 “Taxonomy” to translate these into biological “strategies” that have or could inspire innovation [22]. Other initiatives have combined this functional approach with advanced algorithms, processing large datasets (texts, images, etc.) that support online biomimicry tools, such as Mimicus (https://www.lib.mimic.us/ap/ accessed on 1 January 2024), developed by Sun-Young Kim and colleagues, or PeTal from NASA [23]. Finally, one of the most promising ways to explore scientific literature is the “trade-off” approach, focused on the balance between different functions necessary for a species’ survival [24,25]. Such an approach, combined with biomimetic ontologies and engineering-to-biology thesauri developed during the 2000s [26,27,28], led to the development of the E2BMO tool. This computer-aided user interface allows the exploration of scientific literature using the concepts of functions and trade-offs [29]. All these tools help to generate ideas for bioinspired products, processes, and systems. However, at present, these (and other available resources) orientate the user towards a limited set of taxa. In this article, we present a new tool, Bioinspire-Explore (https://Bioinspire-Explore.mnhn.fr/ accessed on 1 January 2024), developed to simplify access to biodiversity data in order to facilitate “biology push” approaches to bioinspiration without inherent taxonomic bias. Bioinspire–Explore offers the bioinspired designer the phylogenetic “tree of life” as a starting point, allowing the exploration of (known) extant biodiversity in its entirety. It is structured around taxa (i.e., names of species, genus, families, etc.) entered by users based on their curiosity or existing knowledge. While the immensity of taxonomic data may seem overwhelming to a non-biologist [30], Bioinspire-Explore aims to support the user in their discovery of the living world and present connections between taxa and associated information (ecological, climatic, visual) of relevance. Above all, its value lies in its ability to display the semantic proximity between taxa, biological processes, habitats, and physical measurements within the scientific literature on bioinspiration. Bioinspire-Explore does not replace interaction with biologists: transdisciplinary collaboration is essential to understanding the principles that govern the living world [31] and for effective bioinspiration [32,33]. However, Bioinspire-Explore provides a means to begin biological exploration, individually or as a team, and to stimulate bioinspiration in all its forms (i.e., biomimicry, bioassistance, bioproduction, Nature Based Solutions, etc…) [34].

## 2. Materials and Methods

### 2.1. Data Sources

Table 1 presents the data sources behind Bioinspire-Explore. Taxonomic data are provided via the application programming interface (API) of the Global Biodiversity Information Facility GBIF (https://gbif.org/ accessed on 1 January 2023), with names of taxa (vernacular and scientific), their occurrences, (spatial coordinates), taxonomic rank, and images. The GBIF taxon identification number is used to query Wikidata. If the Wikidata node of the taxon in question is found, the first paragraph and image of its Wikipedia page is displayed on the Bioinspire-Explore interface via the Wikipedia representational state transfer (REST) API. Going through the Wikidata graph prevents “false friend-associated” pages (e.g., avoiding a query using the word “Vespa” displaying information on motorcycles). Images are imported through the GBIF API and originate from the iNaturalist database.

Climatic data for each taxon are provided by the WordClim (https://www.worldclim.org/ accessed on 1 January 2024) initiative. Plotting a taxon’s climatic conditions involves several steps: (i) Nine monthly climatic variables (elevation, average precipitation, max. precipitation, min. precipitation, solar irradiance, average temp, min. temp, max. temp, and wind speed) are downloaded. (ii) Taxon occurrence maps are downloaded from GBIF, containing a list of hexagonal geographical zones where the taxon has been recorded. (iii) The climatic variables maps are overlaid with the occurrences maps. (iv) The final metric, as displayed in the Climate Tab, represents a spatial mean over the distribution range, averaged over 12 months of the year. To provide the range, maximums and the minimums are calculated over the monthly distributions.

### 2.2. Architecture

Bioinspire-Explore is an open-source tool developed as a full-stack application; in other words, it comprises an entire set of software/technologies with both front end (user-side) and back end (server-side) elements. The back-end aspects are based on Python scripts and Fast APIs, whereas the front end is written using the JavaScript React application. The full repository of the Bioinspire-Explore project can be found here: GitHub https://github.com/ceebios/Bioinspire-Explore-app (accessed on 1 January 2024).

### 2.3. Word Vectorization and Expansion

Bioinspire-Explore allows the user to evaluate the semantic proximity of entities (“Taxon”, “Biological Process”, “Habitat”, and “Physical Measurement”) within a body of scientific literature pertaining to bioinspiration. This supports bioinspiration by offering potential links between a taxon and its associated biological functions, environment, or physical characteristics. Bioinspire-Explore’s functionality is based on a Word2Vec model [35] that was trained on tokenized sentences from the BIOMIG Corpus. The BIOMIG corpus is a dataset developed by Tchakarov et al. 2023 [36] containing scientific papers evaluated as relevant for bioinspiration.

In the Word2Vec model, entities were defined as follows:“Taxon” entities were taken from the GBIF taxonomic backbone [37].“Biological Process” entities were derived from a Wikidata list of biological processes https://www.wikidata.org/wiki/Property:P682 (accessed on 1 January 2023), filtered manually by the authors to remove irrelevant content.“Habitat” entities are those listed under “biome” in the ontology ENVO [38].“Measurement” entities are listed as the “quantities” in Ontology of units of Measure (OM) https://www.ebi.ac.uk/ols/ontologies/om (accessed on 1 January 2023).

The compatibility score between entities is calculated as follows: for a given word entered by the user (e.g., “vespa”), the model looks for the N possible entries in the model that contain this word (e.g., “vespa mandarina”, “vespa ducalis”). The cosine similarities between the vectors of these entries and the vectors of all the entities stored in the model (e.g., the words “resistance” or “venom”) are calculated. The compatibility of each entity is defined as the minimum of these N similarity values. For each input word, N similarity values are provided, with 100% indicating “perfect compatibility” between the term chosen (in this case, “vespa”) and the associated entity. Bioinspire-Explore’s objective is to suggest entities that are the most compatible with all entries containing the word “vespa”. The final top 20 suggestions shown on the user interface represent those entities with the highest compatibility score.

## 3. Results

### 3.1. Dynamic Taxon Search Capability

Bioinspire-Explore’s first key feature is to support users in their search for a taxon of interest. As text is entered, the search bar automatically suggests up to 20 scientific names associated with a query at different taxonomic levels (i.e., species, genus, family, order, etc.). Each taxonomic rank is colour-coded, from red for species to blue for kingdom.

As seen in Figure 1A, a query beginning with the text “hymenopt” yields ten taxa suggestions at different levels. (Note that an identical request on Google may yield many more results, but these are focused solely on the order Hymenoptera, and not its taxonomic “children”.) Users can search for species using either their Latin (scientific) or common (vernacular) names. For example, the user request “mockingbird” provides eight species names identified as mockingbird taxa, as seen in Figure 1B. Table 2 shows the taxonomic results of sample user queries (e.g., “blue whale”, “mockingbird”, “nikolay”, and “escherichia”), and their explanation.

For a given query or set of queries, an accompanying diagram is created that allows the user to locate their taxa in the “tree of life”. This dynamic taxonomic representation is based on the GBIF taxonomic backbone. For example, let us say a user is interested in exploring the taxonomy of insect species that could inspire the management of heat and light, in this case bees and ants [39,40]. Following the user’s query for the bee *Megachile kalina*, a linear taxonomic diagram is created, presenting its taxonomic “location” in Figure 2A. Subsequently, five other species of the genus *Megachile* were searched for (Figure 2B) with Bioinspire-Explore showing the taxonomic connections between these “new” species and the “original” species *Megachile kalina*. The user then enters a new query, this time for the species of desert ant *Cataglyphis bombycina*. Bioinspire-Explore “remembers” the previous searches (unless the clear button is used in between queries) and adds this species’ taxonomy, leading to a bifurcation in the diagram seen in Figure 2C. These taxonomic diagrams are exportable as a PNG files that the user can then download.

### 3.2. Biological and Environmental Data

A great deal of biological and environmental data are freely available online. However, at present, exploring this information involves going back and forth between different websites, which can be time consuming and distracting. Bioinspire-Explore has been designed as a “one stop shop” connecting multiple external open-access data sources via a taxon’s name.

Figure 3 shows the results of a query for *Arnica montana*, a flowering plant species found in western Europe. The first window displays the first paragraph and first image of the Wikipedia page associated with Wikidata node for the species Q207848 (https://www.wikidata.org/wiki/Q207848 (accessed on 1 January 2023)). If the Wikidata graph is missing a node for the taxon in question, this window will be empty. An external link will then automatically search for the taxon name in Wikipedia to provide associated data for this window. The second window, seen in Figure 3B, shows occurrence data for *Arnica montana*, plotted on a world map (with an option to zoom in and out, depending on the scale of interest). Figure 3C displays iNaturalist images tagged with *Arnica montana*. By clicking on an image, the user will be led to the iNaturalist website. Figure 3D shows the average climatic conditions, as well as minimums and maximums, for *Arnica montana*, as found under the forth window. Windows 3 and 4 aim to provide information on a taxon’s environmental context, highlighting the abiotic conditions to which it is adapted (temperature, rainfall, sun exposure, wind speed, etc.).

### 3.3. Semantic Proximity between Biological Entities

As outlined, Tchakarov et al. 2023 [36] developed a neural network model Word2Vec [35] based on a corpus of scientific articles judged relevant for bioinspiration (the “BIOMIG Corpus”). Using this model, Bioinspire-Explore is able to assess the semantic proximity of text entities (taxon, biological process, habitat, physical measurements) stored as word “embeddings” within this scientific corpus. These entities were chosen as being the most relevant in the context of a “biology push” approach to bioinspiration, providing users with ideas in terms biological processes, functions, structures, and systems of potential interest for innovation (see examples in Table 3).

Figure 4A presents a more detailed example of how these text embeddings, and the evaluation of their semantic proximity with the scientific corpus, work in practice. Here, seven text entities are represented as vectors in a fictive 3D-space: three taxon names (“*Pica pica*”, “*Corvus corone*”, and “*Vespa soror*”); two biological processes (“diving” and “swarming”); and two habitat characteristics (“forest” and “tropical”).

The taxon entities *Pica pica* and *Corvus corone* are represented as close vectors, because both these taxa belonging to the Corvidae family. The Asian wasp *Vespa soror* is represented as a vector with more distant coordinates from the two Corvidae, as their phylogenetic relationship is further apart. The biological process “swarm” is shown to be semantically closer to the *Vespa soror* than to the two Corvidae. This is because swarming is a behaviour typically associated with this wasp (rather than Corvidae), and the scientific literature reflects this. Conversely, “diving” is more likely to be found in Corvidae than *Vespa soror*, and therefore appears closer to the two birds. In terms of habitat characteristics, the entity “tropical” is directionally closer to the text embedding of *Vespa soror* than to the Corvidae, as the latter live in temperate climates of western Europe.

How are the results of this neural network model presented within Bioinspire-Explore? In Figure 4B, we show two screenshots of the “Biodiversity” page within Bioinspire-Explore, where the concept of semantic proximity between taxa, biological processes, habitats, or measurements is explored. In the first screenshot, we explore entities related to our wasp *Vespa soror*. We start with the “taxon” tab to understand how this wasp is connected to other taxa within the scientific literature. As to be expected, the model indicates greatest semantic proximity with other insect families of the order Hymenoptera, such as Platygastridae, Sapygidae, Pergidae, or Argidae. By clicking on the “Process” tab, we find the biological process “swarm” with a 74.5% semantic proximity to *Vespa soror*. Selecting this process, we are interested in understanding the proximity of the “Habitat” entities connected to both *Vespa soror* and swarming behaviour. This is shown in the second screenshot of Figure 4B. Having been asked for habitats semantically close to both *Vespa soror* and swarm, Bioinspire-Explore cites “tropical”, “meadow”, and “desert” as related entities. The compatibility scores between this trio of habitat entities are 67%, 64%, and 63.5%, respectively. This means that, in the scientific literature (BIOMIG Corpus) used by the Bioinspire-Explore model, connections between swarming (process) of *Vespa soror* (taxon) in tropical areas (habitat) appear to be the most significant, with a combined score of 67%.

### 3.4. An Example “User Journey” in Bioinspire-Explore

Figure 5 presents how a user might explore the Bioinspire-Explore tool. If they are new to the concept of bioinspiration, they may start by visiting the “Bioinspirations” page (Figure 5A) that presents a handful of existing initiatives in the field of bioinspired design (using images, videos, text, and references). Via this page, they might read about the navigational abilities of the desert ant *Cataglyphis fortis*. They decide they would like to learn more about this species via the “Biodiversity” page (being redirected automatically via the species name, or by typing its name directly into the search box). Figure 5B shows the information that is provided at this stage (as already described in Figure 2): the species taxonomic position, its appearance, its geographical distribution, and climatic niche.

Finally, the user may wish to dive deeper via the “Go further” page in Bioinspire-Explore (Figure 5C), getting an overview of closely related entities within the scientific literature associated with *Cataglyphis fortis*. These insights usually lead the user either (i) to further research on the taxon in question (using other platforms (EOL, GBIF…) or via the scientific articles presented in the Biomig Corpus) or (ii) to a new request about another taxon presenting a similar interest (for example, the user might identify the desert ant *Melophorus bagoti* as an alternative additional biological model for navigation).

## 4. Discussion

### 4.1. Bioinspire-Explore’s Audience

Bioinspire-Explore is an innovative new tool, providing a unique way to explore biodiversity data and visualise relationships between taxa as well as between entities of relevance to bioinspiration. Bioinspire-Explore’s focus is on a “biology-push” approach to bioinspiration, that is to say, the identification of biological systems of potential interest, the abstraction of their associated functions, followed by their transposition to a technical domain. However, Bioinspire-Explore could also be used to support of a “technology-pull” approach, as described by Fayemi et al. [41], i.e., once a team has transposed technological needs into biological functions, and is searching for biological models that fulfill those functions (See Figure 6).

The authors did not carry out a formalized process to generate scientifically robust user feedback for the first iteration of the Bioinspire-Explore tool. Bioinspire-Explore was, however, thoroughly tested with users from different institutions and communities on five separate occasions (three Masters courses and two scientific workshops) and received constructive feedback that fed into the project, either into the tool itself or this article. Future feedback will be collected via a dedicated email address.

Moreover, although designed to help bioinspiration practitioners, Bioinspire-Explore could also serve as an educational tool in many different settings. Not only does the tool facilitate access to biological data, it offers an interactive interface to “play” with taxonomic information and phylogenetic relationships. Over and above finding relevant data for a bioinspired project, Bioinspire-Explore helps users understand how life on earth is organised scientifically, putting bioinspiration into its biological context. Thus, Bioinspire-Explore could be used as a way to tackle both difficulties in understanding biology as well challenges associated with the transfer of knowledge from biology to technology, as identified by Wanieck et al. [42]: both obstacles within the bioinspiration field.

### 4.2. Challenges and Next Steps

Bioinspire-Explore represents a first iteration in the development of an online “biology push” bioinspiration tool. As a prototype, it still faces challenges in terms of data sources and functionality. With regard to data sources, Bioinspire-Explore provides access to international biodiversity data via the most well recognised open access source: the Global Biodiversity Information Facility (GBIF). Despite the significant advantages associated with facilitating access to this vast dataset, Bioinspire-Explore’s connections with GBIF mean that its limitations are interwoven as well. In other words, Bioinspire-Explore’s data contain the same biases as GBIF’s, and the same challenges in terms of staying up to date with constant phylogenetic and taxonomic developments [43,44,45,46]. For example, if a species’ scientific name has recently changed, Bioinspire-Explore will present only those data relating to the new species’ name and not its previous name. This means that all data associated with its previous name are no longer available to users. The reverse also holds true, if taxonomic updates have not yet been made in GBIF, Bioinspire-Explore will relay nomenclature inconsistencies. Nevertheless, efforts are underway to improve the quality of GBIF data [47,48], and this will directly benefit the Bioinspire-Explore tool and its taxonomic accuracy.

Moreover, not all geographical areas covered by GBIF have the same density of data points [49,50]; nor do all taxonomic groups have equivalent amounts of data on their geographical distribution [20]. This affects the accuracy of climatic ranges of certain taxa calculated within Bioinspire-Explore, for example.

In terms of the bioinspiration literature that informs the analyses of semantic proximity metrics, the “BIOMIG corpus” is updated every three months with new literature relevant to bioinspiration. Although the BIOMIG corpus has been developed to include all pertinent scientific articles (even when they do not mention bioinspiration or biomimetics explicitly), this corpus cannot claim to be exhaustive: there will inevitably be published scientific literature useful to bioinspiration that is not automatically identified as such. Despite the regular updating of the BIOMIG corpus (every three months), the Word2Vec model, trained from this corpus, is static (and does not change). This is a deliberate choice of the development team. Indeed, the point of Word2Vec is to unravel the semantic meaning of entities (words), and the relationship between these entities, using a large number of sentences. Considering the extent of the BIOMIG corpus (several decades worth of scientific papers), a regular re-training of Word2Vec on an updated corpus would not provide significant changes in these semantic relationships between words. For example, the entity “Morpho menelaus” is currently related to “reflection” or to other butterfly taxa; in a year, the small amount of papers related to Morpho will not be a sufficient “signal” to modify this semantic proximity. That being said, we should indeed retrain the Word2Vec model every few years, or could even replace this semantic expansion tool with more costly Large Language models in the future.

Our intention is to continue to improve Bioinspire-Explore’s functionality and scope in the future, based on feedback following its use in educational and professional settings. Several possible avenues for its improvement already exist. From a functionality perspective, building the tool’s capacity for non-taxa entry points is one priority. Presently, Bioinspire-Explore’s capability lies chiefly in the assessment of semantic proximity between a taxon (as the starting point) and other entities (biological processes, physical measures, and habitats) in the scientific literature. For now, starting a search with one of these other entities provides less obvious results in terms of connections in the scientific literature. For example, if one were to begin a search with “energy” as a physical measurement, and then “photo-protection” as a biological process, the results in terms of semantic proximity linking these entities to specific taxa are weak. This currently limits the ability of a bioinspired designer to pursue “property-led” rather than “taxon-led” research.

In terms of climate data, Bioinspire-Explore uses the last available “snaptshot” of the WorldClim database, issued in 2020, and covering the 1970 to 2000 period of time. We are aware of the limitations of this database’s scope and the importance of having up-to-date relationships between taxa and their precise climatic conditions. This aspect could be improved in future iterations of the tool.

In terms of the potential integration of new datasets into Bioinspire-Explore, two types of information stand out: morphological and palaeontological. Morphological descriptions are those used to identify living organisms, presenting their specific structures and characteristics. These properties are closely linked to an organism’s function, and therefore of potential interest to bioinspiration practitioners. Bioinspire-Explore could be expanded to include morphological (and anatomical) knowledge extracted from relevant scientific literature via AI text analysis methods. This information could be presented in the form of graphs or RDF triplets, for example, as proposed by Sahraoui et al. [51].

Finally, palaeo-bioinspiration is a growing field [52], and one that requires access to information about species from previous geological epochs. The biodiversity data available through Bioinspire-Explore are currently based on a classification system that does not fully take into account the fossil record. Although international, open-access palaeobiodiversity databases exist [53], their utilisation would require the harmonious integration of the classification of fossils with that of current species. Presently, integrating the multiple branches of fossil lineages at higher taxonomic levels pose both theoretical and technical challenges; these difficulties would need to be evaluated and resolved before a unified catalogue of past and present biodiversity could be made available through Bioinspire-Explore. The tool is currently being tested by Masters and PhD students as part of their bioinspired design curricula, as well as professionals within the bioinspiration domain. Their user feedback will provide insight on the functional, visual, and ergonomic choices made by Bioinspire-Explore’s development team. Particular attention will be given to improving the semantic expansion tool to assess whether it is understood, intuitive, and useful in the application of bioinspiration.

## 5. Conclusions

In a context of growing environmental challenges, professionals are increasingly looking to the living world to provide inspiration for more sustainable alternatives to today’s systems. Bioinspire-Explore’s principle audience is this new generation of designers, engineers, chemists, architects, urban planners, and more. It can be operated by a single user, but can also facilitate interaction between different disciplines in a group setting. Crucially, Bioinspire-Explore is intended to act as a guide, not as a replacement for the active involvement of biologists in bioinspiration projects [31,54]. Rather, it orientates the user towards promising information regarding living systems of interest and presents those systems in their scientific context. It offers relevant scientific literature to bioinspiration practitioners seeking stimulus for many different variants of bioinspiration [34]: for example, exploring properties of biological materials to support biofabrication/bioproduct engineering initiatives; or identifying ecological characteristics of taxa that could be integrated into nature based solutions. Bioinspire-Explore is a powerful tool for browsing and visualizing biological data, using taxonomy as an entry point. The platform’s user-friendly interface and comprehensive databases make it an accessible way to explore biological data, particularly as part of a “biology push” approach to bioinspiration, and provide meaningful insights into the diversity and evolution of life on earth. Its unique value lies in its ability to combine, within a single tool (i) taxonomic data; (ii) ecological information; and (iii) indications of the semantic proximity of terms (“entities”) cited in bioinspiration-related literature. In the future, we anticipate that Bioinspire-Explore will continue to evolve and improve, with the addition of new features, functionality, and datasets in order to respond to the growing ambitions of the bioinspiration community.

## Figures and Tables

**Figure 1 biomimetics-09-00063-f001:**
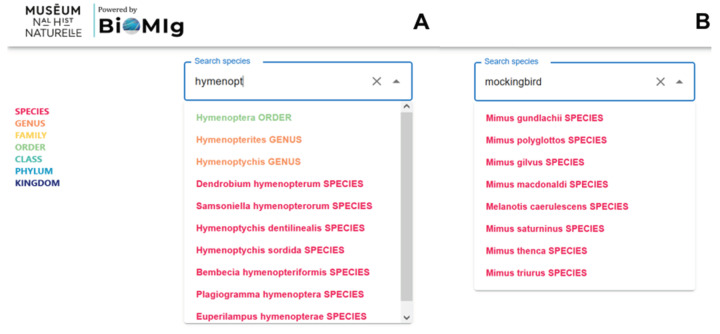
Search bar automatically suggests taxonomic terms associated with a query, covering both (**A**) scientific names and (**B**) vernacular names.

**Figure 2 biomimetics-09-00063-f002:**
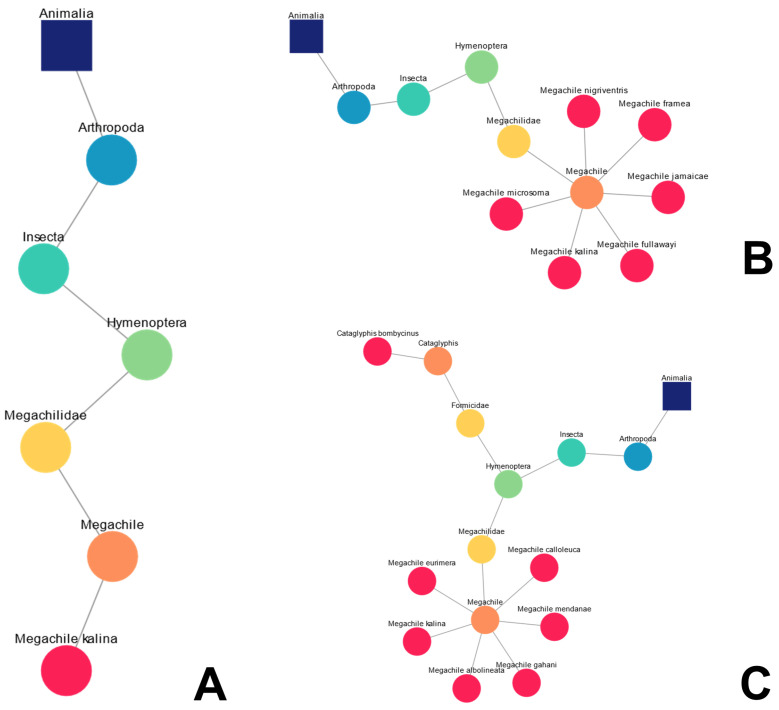
Examples of dynamic taxonomic diagrams following multiple queries associated with bees. (**A**) Linear diagram created after single search for the bee species *Megachile kalina*; (**B**) expansion of the “taxonomic children” of the genus *Megachile*; (**C**) subsequent search for the desert ant species *Cataglyphis bombycinus*.

**Figure 3 biomimetics-09-00063-f003:**
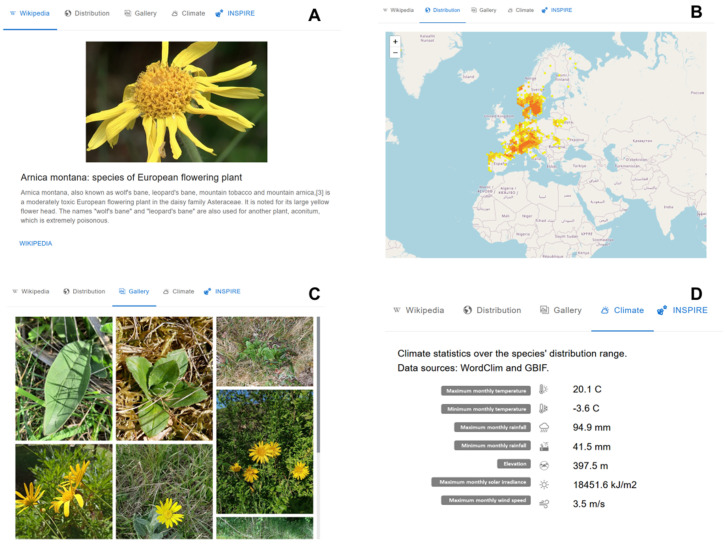
Four main types of information are provided through Bioinspire-Explore for a selected taxon *Arnica montana*. (**A**) Its Wikipedia page, (**B**) spatial distribution, (**C**) images, and (**D**) climate characteristics.

**Figure 4 biomimetics-09-00063-f004:**
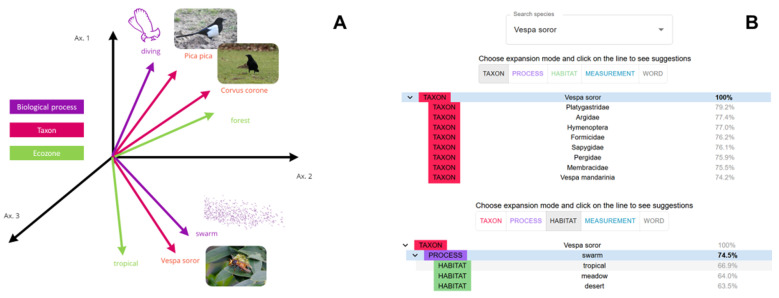
(**A**) Schematic illustration of semantic proximity of various types of entities “taxon”, “ecozone/habitat”, “biological process”; (**B**) screenshot of the semantic expansion module in Bioinspire-Explore. Here, the taxon entity *Vespa soror* appears semantically related to other taxa such as Hymenoptera.

**Figure 5 biomimetics-09-00063-f005:**
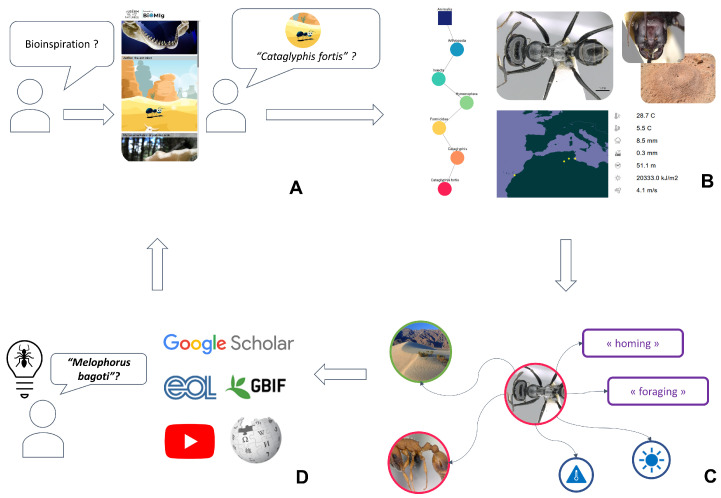
User journey within Bioinspire-Explore, illustrating the functionalities of different tabs within the tool: (**A**) visit of the “Bioinspirations” page presenting bioinspiration examples; (**B**) then moving to the “Biodiversity” page to learn more on a given taxon; (**C**) the “Go Further” page gives an overview of closely-related entities within the scientific literature; and (**D**) end of the user journey by either doing another research or using the pre-filled external links.

**Figure 6 biomimetics-09-00063-f006:**
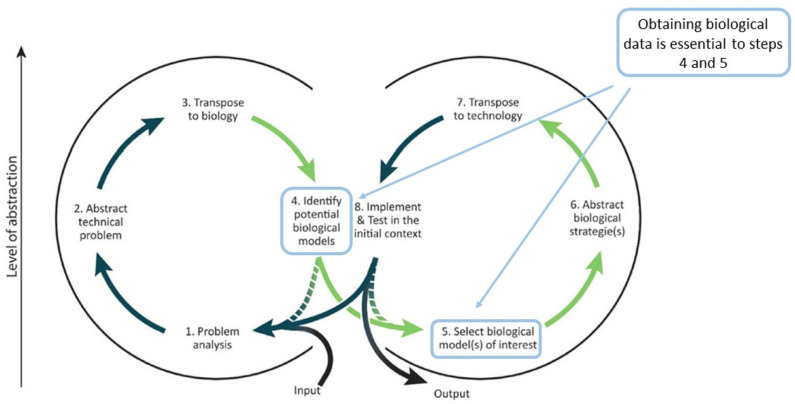
Possible uses Bioinspire-Explore within a biomimetic methodological framework. After Fayemi et al. (2017) [41].

**Table 1 biomimetics-09-00063-t001:** Data sources and their providers.

Data Type	Sourced From
Taxonomic backbone	GBIF
Occurrence data	GBIF
Taxon-related photographs	iNaturalist
General information on given taxon	Wikipedia
Geo-climatic data	WorldClim
List of biological process	Wikidata
Related entities, e.g., co-cited species	Biomig corpus

**Table 2 biomimetics-09-00063-t002:** Illustration of how the search bar helps the user navigate taxonomic data, depending on the type of request.

User Request	Propositions of the Research Bar	Explanation of the Result
“blue whale”	*Balaenoptera musculus*, *Prionace glauca*	two taxa associated with the vernacular name “blue whale”
“mockingbird”	*Mimus gundlachii*, *Mimus polyglottos*, *Mimus macdonaldi*, *Mimus gilvus*, *Melanotis caerulescens*, *Mimus saturninus*, *Mimus triurus*, *Mimus thenca*	eight species associated with the vernacular “mockingbird”
“nikolay”	*Bolinichthys nikolayi*, *Chersodromia nikolayi*, *Melamphaes nikolayi*, *Stauroneis nikolayi*, *Conterinia nikolayi*, *Prodalmanitina nikolayevi*, *Pseudepipona nikolayi*	seven taxa whose specific epithet contain “nikolay”
“escherichia”	*Escherichia* (Genus), *Escherichia hermannii*, *Escherichia vulneris*, *Escherichia fergisonii*, *Escherichia albertii*, *Eschericia marmotae*, *Escherichia ruysiae*, *Escherichia coli*	exact match with genus and generic names

**Table 3 biomimetics-09-00063-t003:** Examples of entities related to two taxa queries *Vespa soror* and *Morpho menelaus*).

User Request	Taxon	Process	Habitat	Measure
*Vespa soror*	*Platygastridae*, *Argidae*, *Theroa zethus*	hunting, swarm, autothysis	plant, tropical, meadow	heat, frequency, length
*Morpho menelaus*	*Greta oto*, *Papilio ulysses*, *Callophrys rubi*	polyphenism, colouration, flying	feather, desert, meadow	angle, heat, hydrophobicity

## Data Availability

The data presented in this study are openly available in GitHub at https://github.com/ceebios/Bioinspire-Explore-app (accessed on 1 January 2024).

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
