# Peer review of "Bioinspire-Explore: Taxonomy-Driven Exploration of Biodiversity Data for Bioinspired Innovation"

_biomimetics, 2024, doi:10.3390/biomimetics9020063_

Round 1

Reviewer 1 Report

Comments and Suggestions for Authors

Despite the evident merits presented in the paper on the "Bioinspire-Explore" tool, it is imperative to highlight some challenges and potential considerations that warrant careful attention. The dependence on up-to-date data is critical for the ongoing effectiveness of the tool, emphasizing the need to implement an efficient mechanism for the continuous updating of biological information. Additionally, usability for different professionals, such as designers, engineers, and urban planners, may pose a challenge, requiring an intuitive interface and functionalities adaptable to diverse needs. The absence of real user feedback in the conclusion of the paper could hinder the practical assessment of the tool's effectiveness, underscoring the importance of actively engaging users to identify areas for improvement. Considering that the tool is intended for a global audience, integrating data and information from different regions and ecosystems may be challenging, necessitating a strategic approach to address variations in data availability and quality on an international scale. While these challenges do not diminish the importance of the work, it is crucial to acknowledge them to ensure the continuous improvement and effectiveness of the tool in the field of bioinspired innovation.

Reviewer 2 Report

Comments and Suggestions for Authors

The paper "Bioinspire-Explore: Taxonomy-driven Exploration of Biodiversity Data for Bioinspired Innovation" is well written and aims to be an indispensable resource in bioinspired design, highlighting its potential as an educational tool and discussing challenges and future improvements.

However, the authors should focus more on several potential drawbacks of the Bioinspire-Explore tool which are mentioned in the text:

According to author Bioinspire-Explore relies on the GBIF)for its data. Consequently, it inherits the limitations of GBIF, including biases and challenges in staying current with phylogenetic and taxonomic developments. For instance, if a species' scientific name changes, Bioinspire-Explore will only present data relating to the new name, not the previous one​​. However, I tested the tool, and it showed me, in a few instances, the opposite problem – a lack of changes in taxonomy.

There are inconsistencies in the density of data points across different geographical areas and taxonomic groups in the GBIF database. This variability affects the accuracy of calculated climatic ranges for certain taxa within Bioinspire-Explore​​. Similarly, in my tests, it showed the species name etc. but e.g. no distribution (which was mentioned in the text).

The BIOMIG corpus, which informs the semantic proximity metrics analysis in Bioinspire-Explore, is updated every three months with new bioinspiration-relevant literature. However, this corpus is not exhaustive, meaning that some pertinent scientific articles may not be automatically identified and included​​.

Additionally, there is additional space in line 34 and 53.

Considering the relevance, depth, and innovation presented in this paper, I would recommend its publication in the journal. The insights and functionalities offered by Bioinspire-Explore represent a significant step forward in the field of bioinspiration and could inspire further research and development.

Reviewer 3 Report

Comments and Suggestions for Authors This work is suitable for publication in its current state.   My observations are:   This work is innovative, and facilitates the search for biological information, since it concentrates it in a single search engine. Likewise, it contributes to awakening bioinspiration in users, enhancing the use of biological data in the development of ideas and projects. However, this requires making some adjustments to your webpage (https://bioinspire-explore.mnhn.fr/):   1) On your page https://bioinspire-explore.mnhn.fr/learn, the names "Tramates hirsuta; Tramates versicolor" need to be corrected, the correct thing is "Trametes hirsuta; Trametes versicolor".   2) On the page https://bioinspire-explore.mnhn.fr/explore, When you do a search and use the clear button, it does not completely erase the previous search, it keeps the data in the "section "Wikipedia".  

With these small corrections I consider that the page would be functional.
